# A New Index For The Wintertime Southern Hemispheric Split Jet

Stella Babian[1], Jens Grieger[1], and Ulrich Cubasch[1]

[1]Meteorological Institute, Freie Universität Berlin, Carl-Heinrich-Becker-Weg 6-10, 12165 Berlin, Germany

*Correspondence to:* Stella Babian (stella.babian@met.fu-berlin.de)

**Abstract.** One of the most prominent asymmetric features of the southern hemispheric (SH) circulation is the split jet over Australia and New Zealand in Austral winter. Previous studies have developed indices to detect the degree to which the upper-level mid-latitude westerlies are split and investigated the relationship between split events and the low-frequency teleconnection patterns viz. the Antarctic Oscillation (AAO) and the El Niño-Southern Oscillation (ENSO). As the results were inconsistent, the relationship between the wintertime SH split jet and the climate variability indices remains unresolved and is the focus of this study.

Until now, all split indices definitions were based on the specific region where the split jet is recognizable. We consider the split jet as hemispheric rather than a regional feature and propose a new, hemispherical index that is based on the principal components (PC) of the zonal wind field for the SH winter. A linear combination of PC2 and PC3 of the anomalous monthly (JAS) zonal wind is used to identify split jet conditions.

In a subsequent correlation analysis, our newly defined index (PSI) indicates a strong coherence with the AAO. However, this significant relationship is unstable over the analysis period – during the 1980s the AAO amplitude was higher than the PSI and vice versa in the 1990s. It is probable that the PSI, as well as the AAO, underlie low-frequency variability on the decadal to centennial time scales, but the analysed period is too short to draw these conclusions. A regression analysis with the Multivariate ENSO Index points to a nonlinear relationship between PSI and ENSO i.e. split jets occur during strong both positive and negative phases of ENSO but rarely under "normal" conditions. The PSA patterns, defined as the second and third mode of the geopotential height variability at 500 hPa, correlate poorly with the PSI ($r_{PSA-1} \approx 0.2$ and $r_{PSA-2} = 0.06$), but significantly with the individual components (PCs) of the PSI, revealing an indirect influence on the SH split jet variability.

Our study suggests that the wintertime SH split jet is strongly associated with the AAO, while ENSO is to a lesser extent connected to the PSI. We conclude that a positive AAO phase, as well as both flavors of ENSO and the PSA-1 pattern produce favorable conditions for a SH split event.

## 1 Introduction

The circulation in the Southern Hemisphere (SH) is generally more zonally symmetric than its Northern Hemisphere counterpart, but there are significant zonal variations in the upper-tropospheric time-mean flow. A unique asymmetric feature of the SH winter circulation is the climatological split jet over Australia and New Zealand (Fig. 1).

The split is composed of two distinguishable branches: the northern branch, the subtropical jet (STJ) is located over the South Indian Ocean extending eastwards to the South Pacific Ocean between the latitudes of 25° and 30° S. The STJ is paralleled by the weaker Polar Front Jet (PFJ), which lies over the South Pacific Ocean upto ca. 60° S. A distinct feature of the split structure is a pronounced "gap" between these two branches, characterizing a zone of weak upper-level westerly winds over
New Zealand (Bals-Elsholz et al., 2001).

Several past studies have addressed the existence, location and variability of the wintertime SH split jet. An early study by Taljaard (1972) has linked the existence and strength of the split jet to the outflow from the Asiatic monsoon anticyclone of the Northern Hemisphere. Other studies have variously suggested that the split jet is associated with cold air outbreaks (Mo et al., 1987), breakdown of the Antarctic polar vortex in late spring (Mechoso et al., 1988), phases of ENSO (Karoly, 1989; Chen
et al., 1996), and the AAO (Yang and Chang, 2006).

The AAO, which is in literature also referred to as Southern Annular Mode (SAM), is the dominant climate mode of the extra-tropical SH circulation variability and describes the out-of-phase pressure anomalies in polar and mid-latitude regions (e.g. Lorenz and Hartmann (2001), Thompson and Wallace (2000)). Positive (negative) phases of the AAO are linked to a poleward (equatorward) shift and strengthening (weakening) of the PFJ. The AAO was found to be the main modulator of the
PFJ strength and location (Limpasuvan and Hartmann, 1999) and is thus expected to play a major role in the formation of the split jet.

The ENSO has an impact on both the STJ and the PFJ by directly affecting the Hadley circulation: while El Niño events are linked to increased (equatorial) convection and the PFJ, La Niña phases are marked by a stronger (weaker) PFJ (STJ) (Karoly, 1989; Chen et al., 1996; Kitoh, 1994). Gallego et al. (2005) have confirmed that the ENSO impact on the PFJ attributes
(strength, wavenumber, average latitude) is mainly confined to the Pacific sector where the split jet is located.

Bals-Elsholz et al. (2001) have developed a vorticity based split index and investigated its relationship to both ENSO and the AAO. Their study reveals that the existence of the split jet is dependent upon the presence of the PFJ branch as the STJ is a quasi-permanent feature of the SH winter circulation. The PFJ branch of the split index is indeed correlated with the AAO, but the STJ branch shows no correlation. Thus, overall their index correlates weakly with the AAO. A subsequent study (Inatsu
and Hoskins, 2006) suggested a split index based on zonal mean of zonal wind variations; its correlation with the AAO is significant but low (r = 0.43). Both studies also investigated the relationship between the split jet and the ENSO; neither the index from Bals-Elsholz et al. (2001), nor the one developed by Inatsu and Hoskins (2006) correlated on a significant level with the Southern Oscillation Index (SOI).

The ongoing debate about the relationships between AAO, ENSO and the SH split jet in the published studies raise the
question if an index could be defined that clarifies the relationships between the SH split jet and the large-scale teleconnection indices viz. ENSO and AAO?

In previous studies (Bals-Elsholz et al. (2001) (hereafter BE), Inatsu and Hoskins (2006) (IH)) and in the work of Yang and Chang (2006) (YC), indices were developed to detect the characteristics of the SH climatological split in the Austral winter.

These indices represent the split structure in several meteorological variables and at different but comparable levels: 200
hPa vorticity (BE), 300 hPa zonal wind anomalies (YC), 200 – 300 hPa difference in the zonal mean zonal wind (IH). The

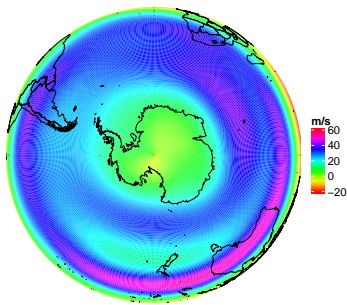

**Figure 1.** Climatological zonal wind (m/s) in 200 hPa in Austral winter (JAS) averaged for the period 1979–2015.

similarity in the construction of all these split indices is a definition based on the regions accounting for the two branches of the split jet (STJ, PFJ) and the gap between them over the Australian / New Zealand region. Although these indices are similarly constructed, the respective studies revealed inconsistent relationships with the AAO and ENSO.

Additionally, there remains a lack of clarity about the link between split jet variability and the Pacific South American (PSA) patterns, although the latter were found to be associated with ENSO and the AAO teleconnection indices. The PSA patterns are conventionally seen as Rossby wave trains emanating in quadrature to each other from the tropical Pacific towards Argentina and serve simultaneously as waveguide for eddies moving south. On interannual time scales the PSA-1 mode is tied to ENSO, while the PSA-2 pattern is associated with the quasi-biennial component of ENSO (e.g., Mo, 2000). Furthermore, although called an "annular" mode, the AAO contains asymmetries, which are most pronounced in Austral winter and over the Pacific sector. This (tropically forced) component of the AAO is related to a fixed active Rossby wave source and resembles the spatial structure of the PSA patterns (Ding et al., 2012).

The lack of knowledge about the relationship between the split jet variability and the PSA patterns, and the inconsistencies about its connection to ENSO and the AAO have motivated this study to develop an improved SH split jet index. The split jet is one of the most important features of the interannual variability in the SH winter circulation and its pattern is not only centered over the Australian / New Zealand region but bears also hemispheric signatures (Yang and Chang, 2006). Consequently, we assume that the split jet variability is already intrinsically contained in one or several leading modes of the SH winter zonal wind field.

In this study, we provide a PC based methodology to describe the SH wintertime split jet to clarify the relationship between the SH split jet and the large-scale teleconnection indices (ENSO and AAO). This paper is organized as follows: the data set and methodology used for the reconstruction of the split jet indices and for the calculation of the new PC based split index (PSI) are introduced in the data and methods Section 2. The definition of the new index is proposed in Section 3. We will use the PSI to examine the relationships between split phases and the known large-scale variability modes (ENSO, AAO and PSA) in Section 4. Finally, the main results are summarized and discussed in Section 5.

## 2 Data and methods

### 2.1 Data

This study uses the ERA-Interim (Dee et al., 2011) reanalysis data sets from the European Centre for Medium Range Weather Forecast (ECMWF). ERA-Interim was selected because ECMWF products (e.g. MSLP and geopotential height at 500 hPa from ERA-Interim) were found to be the most reliable, in particular over the Antarctic continent (Bracegirdle and Marshall, 2012).

We use monthly zonal wind and geopotential height data covering the winter seasons, defined here for the months from July to September (JAS) in accordance with the three previous studies (Bals-Elsholz et al. (2001); Inatsu and Hoskins (2006); Yang and Chang (2006)). All fields were used on a 0.75° grid and were analyzed for the period between 1979–2015.

The monthly resolved teleconnection indices used in this study, namely the Antarctic Oscillation Index and the Multivariate ENSO Index, are both freely available from NOAA's website (http://www.noaa.gov/). All time series used in this study are standardized so that they have a mean of zero and a standard deviation of one.

### 2.2 Empirical Orthogonal Function (EOF) Analysis

Analysis of atmospheric circulation patterns can be done by means of an Empirical Orthogonal Functions Analysis (EOF), which is also referred to as Principal Component Analysis (PCA) (Jolliffe, 2002; Hannachi et al., 2007). By definition, an EOF analysis reduces a data set containing a large number of variables to a data set containing fewer new variables, which are linear combinations of the original ones (Wilks, 2011). The first principal component (PC) is the linear combination with the largest variance. To analyze the variability of the SH zonal wind, we first removed the seasonal cycle by taking the anomalies with respect to a mean annual cycle, which in turn was obtained by an average of the individual winter months of the year.

The individual grid cells are then centered and scaled with the square-root of their latitude to account for different grid cell sizes. We then performed an EOF analysis of the 111 winter months for the period between 1979–2015. The associated PC gives the respective time series. By definition, the PSA modes are the second and third EOF of the 500 hPa geopotential height anomalies over the SH (Mo, 2000). The associated PCs give the PSA-1 (second PC) and PSA-2 (third PC) time series.

### 2.3 Composite and Correlation Analysis

25 Composites of months with anomalously high or low split index are the basis for investigating the potential mechanisms associated with splits in the time-mean SH winter circulation. Monthly values exceeding (below) the respective normalized index mean (1979–2015) by about plus (minus) one standard deviation (Hendon et al., 2007) were averaged and defined as positive (negative) composites. The Pearson's correlation coefficient was calculated to quantify the linear relationship between the split jet time series and several indices of the known large-scale oscillations (i.e. AAO and ENSO) (Wilks, 2011).

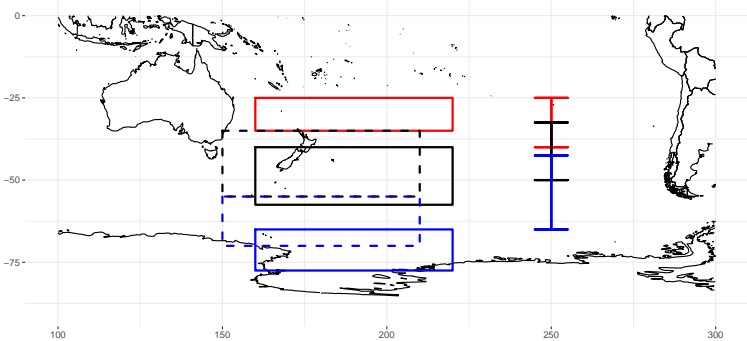

**Figure 2.** Area definitions of three earlier studies. Red lines show the STJ part, blue lines the PFJ part and the black lines the gap between the two branches defined by the different studies. Continuous lines refer to the SFI of Bals-Elsholz et al. (2001) and dashed lines mark the two regions defined in Yang and Chang (2006). The right side shows the meridional boundaries used in the split index defined by Inatsu and Hoskins (2006).

## 2.4 Split Jet Index Definitions

Figure 1 shows the 200 hPa climatological SH winter (JAS) zonal mean wind and the typical time-mean split jet composed of a subtropical (at 30° S) and a polar branch (roughly at 60° S) as well as the intervening characteristic zonal wind minima.

The three split jet indices which were introduced in the Section 1 are based on flow characteristics over the specific regions of Australia and New Zealand (see Fig. 2). Each index is based on a Subtropical Jet (STJ) and a Polar Front Jet (PFJ) component and a gap between these two branches, except for the YC index which lacks the subtropical part. Table 1 provides information about these areas and gives an overview of the particular data sets used in the split jet references.

### 2.4.1 Split-flow Index (SFI)

Bals-Elsholz et al. (2001) developed the very first split index to evaluate the structure and evolution of the SH split jet structure. A Split-Flow Index (SFI) based on the relative vorticity ($\zeta$) in 200 hPa in three neighboring regions (Fig. 2) was designed by them as follows:

$$SFI = \zeta_{PFJ} + \zeta_{STJ} - \zeta_{GAP} \tag{1}$$

The normalized monthly index gives large negative (positive) values for split (non-split) years. Split flow regimes similar to the climatological mean have normalized SFI values near zero.

| | SFI (BE) | STJ (YC) | STJ (IH) | PSI |
|---|---|---|---|---|
| Reference | Bals-Elsholz et al. (2001) | Yang and Chang (2006) | Inatsu and Hoskins (2006) | Defined within this study (Sec. 3.2) |
| **Data sets** | | | | |
| Variable | relative vorticity | zonal wind | zonal mean wind | zonal wind |
| Level | 200 hPa | 300 hPa | mean of 200 and 300 hPa | 200 hPa |
| Data Source | NCEP NCAR | ERA15 | ERA40 | ERA-Interim |
| Period | 1958 – 2000 | 1979 – 1993 | 1979 – 2001 | 1979–2015 |
| **Regions used for definitions** | | | | |
| STJ | 25 – 35° S | – | 25 – 40° S | |
| GAP | 40 – 57.5° S | 35 – 55° S | 32.5 – 50° S | SH |
| PFJ | 65 – 77.5° S | 55 – 70° S | 42.5 – 65° S | |

**Table 1.** List of split jet definition constraints as suggested in three earlier studies. The indices resemble specific areas depicting the equatorward branch of the split (STJ), the gap between the jets (GAP) and the Polar Front Jet part (PFJ).

### 2.4.2 Normalized Monthly Split-Jet Index (NMSJI)

The Yang and Chang (2006) Split Jet Index (SJI) is based on the difference at 300 hPa time-mean zonal wind anomalies ($U$) in two adjoining areas (PFJ, GAP). The index is subsequently normalized by subtracting the climatological mean and dividing by its standard deviation.

$$SJI = \overline{U}_a^{PFJ} - \overline{U}_a^{GAP} \tag{2}$$

$$NMSJI = \frac{MSJI - \overline{MSJI}}{\sigma} \tag{3}$$

### 2.4.3 Split Jet Index (SJI)

The Split Jet Index published by Inatsu and Hoskins (2006) was defined as the difference between the 200 hPa and 300 hPa zonal mean zonal wind ($U$), between the (overlapping) latitudinal boundaries given in Table 1 and illustrated in Fig. 2.

$$SJI = U_{STJ} - 2 \cdot U_{GAP} + U_{PFJ} \tag{4}$$

In agreement with the weighting of the wind minimum between the jets by 2, the index values rose for both, a strong STJ and/or PFJ and reduce if there is a strong single jet centered in the (gap) region between the two jets.

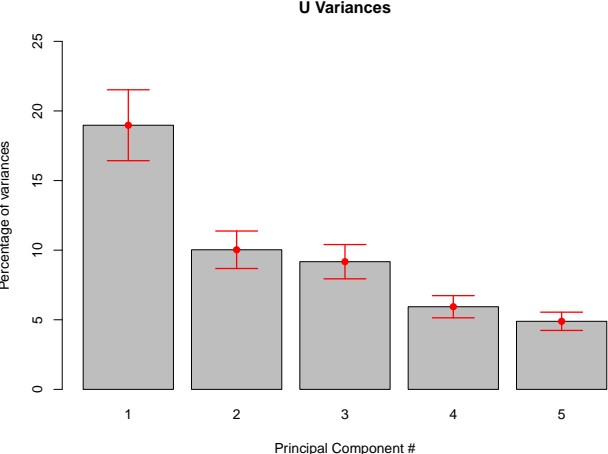

**Figure 3.** Proportions of the total variance [%] associated with the leading 5 EOFs for the 200 hPa zonal wind field of ERA-Interim reanalysis. Error bars are estimates of sampling errors in EOF computation according to "North's rule of thumb" (North et al., 1982).

## 3   A new index for the wintertime SH split jet

As mentioned in Section 1, the wintertime split jet is the most prominent asymmetric feature of the mid-latitude SH circulation centered over the Pacific sector although it bears hemispheric signatures as well (Yang and Chang, 2006). In order to design a PC based split jet index, the three split jet indices defined earlier (Section 2.4) were reproduced, and their respective statistical

5  relationships to the leading modes of the SH wintertime circulation (as depicted by zonal wind anomalies at 200 hPa) have been investigated in this section. The EOF modes showing the largest coherence with the split jet indices (as defined by the respective studies) are assumed to contain the main signals associated with the split variability.

### 3.1   EOFs in SH zonal wind

The three leading spatial patterns of variability (EOFs) for $U$ 200 hPa anomalies on the SH (0–90° S) account for roughly

10  19 %, 10 % and 9 % respectively of the total variability of the 200 hPa zonal wind field (Fig. 3). The 200 hPa pressure level was chosen due to the mechanisms associated with the jet(s) variability, which have their maximum at this level (e.g., Galvin, 2007). The subsequent EOFs (EOF4 and EOF5) represent about 6 % and less of the total variability and are not distinguishable from each other after "North's rule of thumb" (North et al., 1982).

   Worth mentioning, the second and third EOF are effectively degenerate, which means that one of the eigenvalues uncertainty

15  is larger than the spacing between the eigenvalues. By construction, any linear combination of the two EOFs (which describe respective equal amounts of the total variance) will explain the same amount of variance. With the aid of linear algebra, it can be shown that any orthogonal pair of such linear combinations is equally well qualified to be an EOF (Sahai et al., 2014).

|  | PC1 | PC2 | PC3 | PC4 | PC5 | PSI |
|---|---|---|---|---|---|---|
| **JAS – correlation values** | | | | | | |
| **-BE** | **-0.28** | **0.44** | **-0.36** | -0.09 | -0.15 | **0.56** |
| **YC** | -0.24 | **0.56** | **-0.48** | -0.13 | -0.09 | **0.73** |
| **IH** | **-0.49** | **0.57** | **-0.43** | **0.26** | 0.09 | **0.71** |
| **Mean** | **-0.38** | **0.59** | **-0.48** | 0.02 | -0.05 | **0.76** |

**Table 2.** Monthly (JAS) Pearson correlation coefficients of individual leading 5 PCs in the 200 hPa zonal wind as well as the PSI with the split jet indices introduced in the methods section (Sec. 2). By construction, the Bals-Elsholz et al. (2001) index becomes negative during split events and its sign is therefore reversed. Bold values are significant at the $\alpha$ = 1 % level.

The correlation between the leading 5 PCs and the split indices introduced earlier is shown in Table 2. PC1 correlates significantly (r = -0.49) with the zonal mean zonal wind based index developed by Inatsu and Hoskins (2006) and weakly but significantly with the relative vorticity based index designed by Bals-Elsholz et al. (2001). The weaker correlation with the Yang and Chang (2006) index, which is also defined in zonal wind, is because SJI lacks the STJ branch.

Whereas the fourth and fifth PC correlate poorly with any of the indices, the second and third PC show robust relations to split events. While PC2 is strongly positively correlated with the split indices with coefficient values ranging from r = 0.44 (BE) to r = 0.57 (IH), PC3 is negatively related with the split regimes. Pearson's correlation values of PC3 increase from r = -0.36 (BE) to r = -0.48 (YC). Generally, the correlation values associated with the relative vorticity based index designed by Bals-Elsholz et al. (2001) show the weakest links to PC2 and PC3 of the zonal wind field, but are nevertheless significant at

the $\alpha$ = 1 % level. The two zonal wind-based indices (IH and YC) produce the strongest correlation to PC2 and PC3, which is exceeded only by the mean of all three earlier split indices for PC2 (r = 0.59). Altogether, the splits in the westerlies show weaker correlation with PC1, but show an improved association with the higher order PCs in the 200 hPa zonal wind field.

### 3.2  Definition of the PC based Split Index (PSI)

From the correlation analysis of the PCs with the three split indices discussed in Section 3.1, it is apparent that PC2 and PC3 are

associated with the SH wintertime split jet events and resemble the partly split variability. In order to develop a PC based split jet index, PC2 and PC3 which correlate well with the earlier split jet indices are linearly coupled. Consequently, the monthly wintertime (JAS) PC based SH Split Jet Index (PSI) is defined as follows:

$$PSI = PC2_{U_{200hPa}} - PC3_{U_{200hPa}} \tag{5}$$

The PSI is based on the second and third PC of the anomalous 200 hPa zonal wind field over the whole SH during Austral

winter (JAS). The PSI time series, as well as the split indices of the previous studies are displayed in Fig. 4.

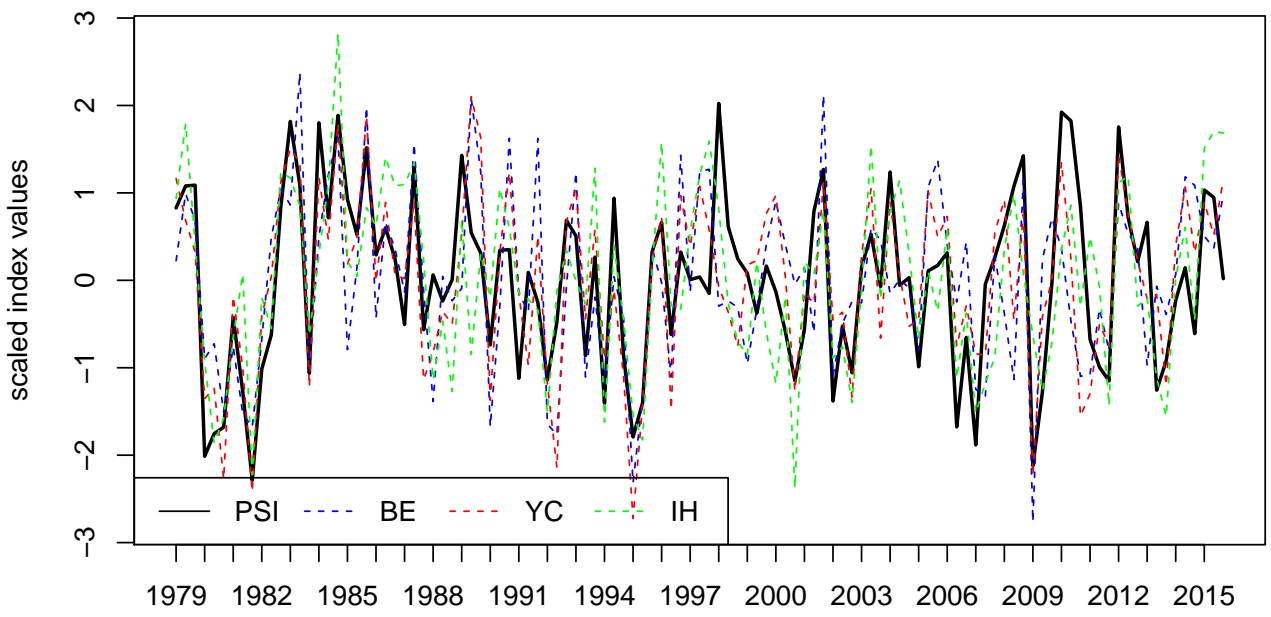

**Figure 4.** PSI (PC2-PC3) index in 200 hPa zonal wind and three earlier split jet indices defined by Bals-Elsholz et al. (2001) (BE), Yang and Chang (2006) (YC) and Inatsu and Hoskins (2006) (IH). All time series were scaled for illustration purposes.

The correlation values between these indices and the PSI at 200 hPa are shown in the last column of Table 2. All indices are well correlated to the PSI with coefficients ranging from 0.56 (BE) to 0.73 (YC) and the mean of all the three earlier split indices raises Pearson's correlation coefficient to r = 0.76. This relationship is significant at the $\alpha = 1$ % level and shows that the linear combination of PC2 and PC3 exceeds values obtained from the performance of the individual PCs (Table 2).

5 ## 4 Results

### 4.1 Composite Analysis of $U$ 200 hPa with PSI

Figure 5 shows the spatial patterns of $U$ at 200 hPa for high (18 cases) and low (19 cases) PSI index phases. The positive composite affirms the characteristic split structure with an existing, but weak STJ, a pronounced PFJ, and a well-defined minima of zonal wind strength in between. The split composite shows that the PFJ strength is not limited to the Australian

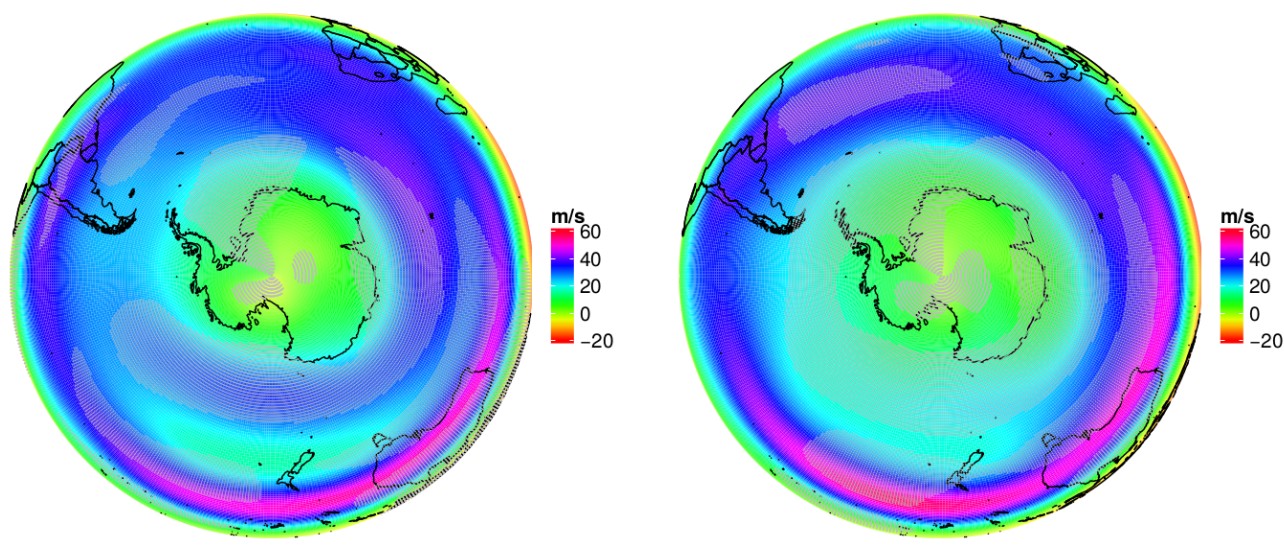

**Figure 5.** Positive (left) and negative (right) composite of the PSI (PC2-PC3) index in 200 hPa zonal wind. Gray dotted areas are significant at the $\alpha = 1$ % level.

sector of the east Pacific, but is enhanced over the Indian Ocean and over the full width of the South Pacific Ocean. The low index composite displays a clear non-split or "merged" jet state with a strong STJ over the Australian continent and a missing PFJ. The spatial patterns of the split and non-split cases are captured well by the PSI and show results similar to the composites of the indices defined in Section 2.4 (not shown).

5      To understand the independent contribution of both components of PSI, i.e. PC2 and PC3, a composite study of both PCs was also done separately. Figure 6 shows the appropriate composites of the second and third EOF in $U$ 200 hPa. The positive composite of PC2 shows the typical split (left column) which developed over the Indian Ocean and strengthened in the western Pacific sector where a clear double jet structure with a pronounced minimum in $U$ 200 hPa between the two jets is visible. The negative composite of PC2 displays a single jet formation (right column). The bottom row of Fig. 6 gives the equivalent

10    composites of PC3. The most striking difference in the split composites (Fig. 6, top left) is the PFJ strength over the western Pacific Ocean and the Drake Passage, which is absent in the PC2 composite.

     Additionally, the positive composite of PC3 shows an established double jet structure over the Atlantic Ocean, which breaks down south of South Africa and results in a strong STJ over the Australian / New Zealand region again (Fig. 6, bottom left). A further feature of the positive composite is the lack of annular structure compared to PC2. In literature, this known zonal wave

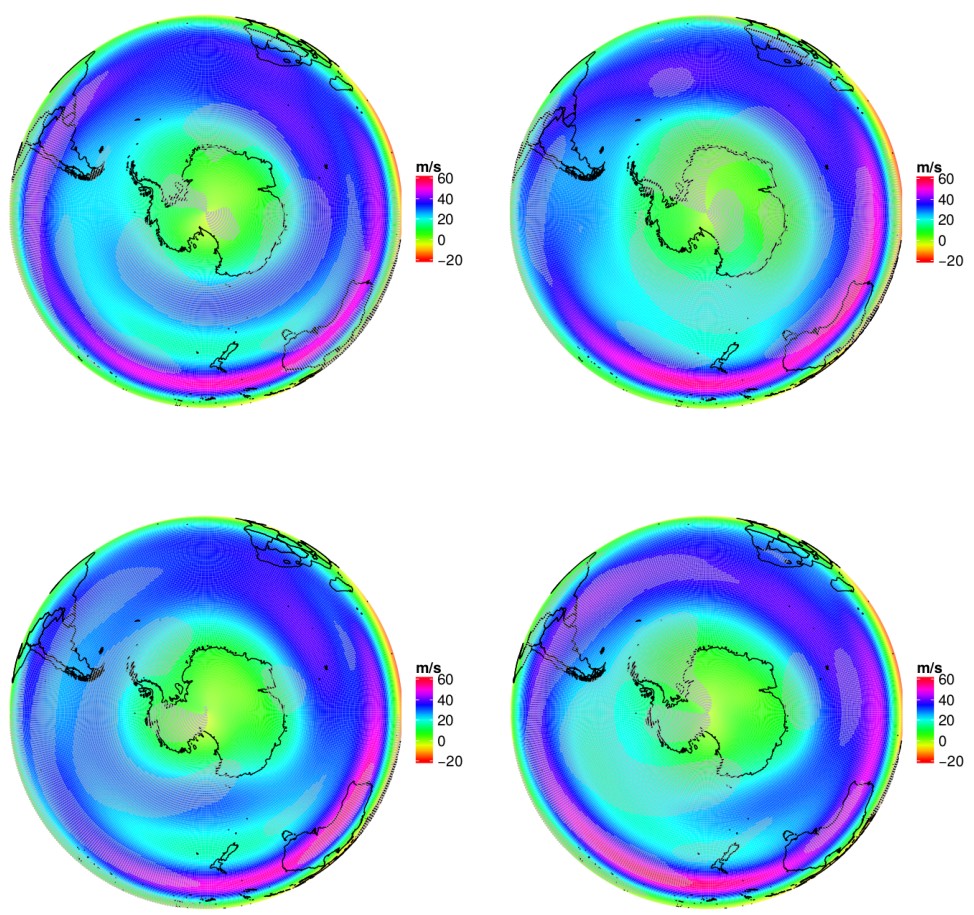

**Figure 6.** Composites of positive (left) and negative (right) phases of PC2 (first summand of Equation 5; top row) and -PC3 (second summand of Equation 5; bottom row) time series of the SH zonal wind in 200 hPa. Please note that the plots are arranged in such a way, that the left column refers to the split condition and the right column to the non-split phase. Gray dotted areas are significant at the $\alpha = 1$ % level.

| | PSI | PC2 | PC3 | BE | YC | IH | Mean |
|---|---|---|---|---|---|---|---|
| **JAS – correlation values** | | | | | | | |
| **AAO** | **0.81** | **0.65** | **-0.50** | **0.45** | **0.66** | **0.58** | **0.64** |
| **MEI** | -0.09 | 0.07 | 0.20 | 0.20 | 0.12 | **0.34** | **0.25** |
| **PSA-1** | 0.19 | **-0.26** | **-0.55** | 0.14 | 0.16 | -0.17 | 0.05 |
| **PSA-2** | 0.06 | 0.20 | 0.13 | 0.22 | 0.20 | -0.12 | 0.11 |

**Table 3.** Monthly (JAS) Pearson correlation coefficients of the three split indices introduced in the methods section (Section 2.4), as well as the PSI with three major SH climate mode indices for the Antarctic Oscillation (AAO), ENSO (Multivariate ENSO Index) and the Pacific South American (PSA) patterns (PSA indices as defined in Mo (2000) for PSA-1 and PSA-2). Bold values are significant at the $\alpha = 1\ \%$ level.

number 3 pattern is often referred to as a modulator of e.g. blocking events in the New Zealand region, and Australian rainfall (Trenberth and Mo, 1985; Pook et al., 2013).

The negative composite (Fig. 6, bottom right) of the second term of Equation 5, i.e. -PC3 (non-split / single jet regime) bears resemblance to the appropriate composite of PC2 but describes a stronger STJ over the central and east South Pacific Ocean and South America. Additionally, the PFJ is weakened in the non-split composite, but south of the Australian / New Zealand region there are still traces of a PFJ.

Summarising, the positive PSI composite (Fig. 5, left) benefits from the combination of the double jet structure with a pronounced minimum between the jets associated with PC2(+) and the strong PFJ over the South Pacific Ocean associated with the "non-annular" component localized over the South Pacific Ocean of PC3(-).

## 4.2 Links between PSI and large-scale climate modes (AAO, ENSO and PSA)

In order to determine whether the PSI is modulated by large-scale phenomena, i.e. the AAO and the ENSO, correlations between the PSI and the teleconnection indices were computed. The respective time series are illustrated in Fig. 7 and the Pearson's correlation coefficients, based on 111 winter months, are summarized in Table 3.

The AAO, also referred to in literature as the Southern Annular Mode (SAM), is defined here as the anomalous geopotential height at 700 hPa (NOAA's AAO) and describes the out-of-phase pressure anomalies in polar and mid-latitude regions (e.g. Lorenz and Hartmann (2001), Thompson and Wallace (2000)). The correlation analysis reveals a strong connection (r = 0.81) between the PSI (defined at 200 hPa) and the AAO. The highly significant correlation accounts for the large influence of the AAO on the PFJ variability, which in turn is associated with the split jet (Fyfe (2003), Gallego et al. (2005)).

Although PSI (split jet index) is significantly correlated with the AAO indicating a strong link with the PSI variability, the PSI has no such relationship with the Multivariate ENSO Index (MEI). Figure 8 shows a regression of the monthly MEI values with respect to the PSI value. The correlation between the PSI and MEI is low (r = 0.04), but by comparing the top left and the top right quadrant of the scatter plot (Fig. 8) to the respective bottom quadrants it is evident that warm (red dots) and cold (red

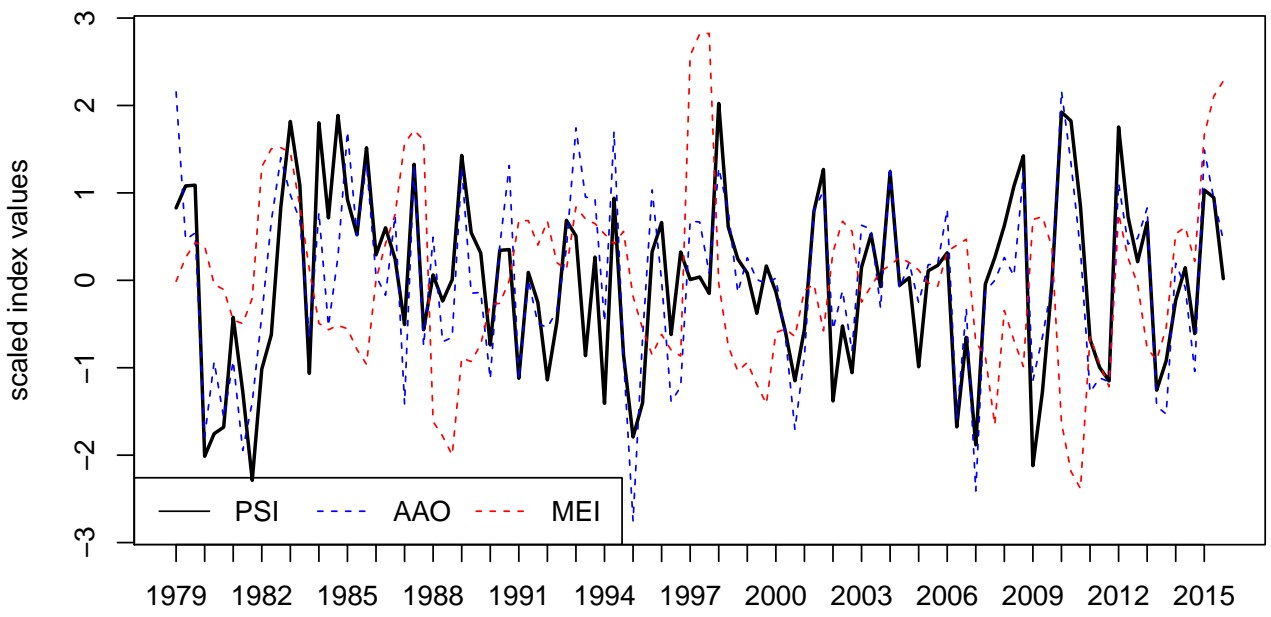

**Figure 7.** Time series of PSI (black solid line), NOAA's AAO (blue dotted line) and NOAA's MEI (red dotted line) from 1979 to 2015.

dots) ENSO events are both associated with positive (or neutral) values of the PSI. Altogether from 24 warm and cold ENSO events that occurred over 111 winter months, it is clear that there was no month with a negative PSI value. This asymmetric and nonlinear behavior damps the correlation coefficient.

Table 4 gives counts of positive (first column), negative (second column) and neutral (last column) PSI months and the corresponding predominant AAO and ENSO states. The box, which emerges when concentrating on the first six lines and columns of the same table, contains all possible $AAO^+/PSI^+$ and $AAO^-/PSI^-$ cases during the analysis period. Although the AAO index is well correlated with the PSI, the last column of the same table shows that 5 $AAO^+$ months are marked as neutral PSI months, and there are 5 months of $PSI^+$ and neutral AAO conditions. Consequently, the jet is able to split under neutral AAO conditions, and not every $AAO^+$ month necessarily leads to a split event. The negative phases of the AAO nearly mirror the conditions of its positive counterpart. Altogether 7 events occurred under $AAO^-/PSI^0$, likewise, 6 months were counted as $AAO^0/PSI^-$ cases.

By revisiting these points in the investigated period between 1979 and 2015, it turns out that 3 months of the first case ($AAO^+/PSI^0$) occurred in a period during the 1980s and 3 of the latter cases ($AAO^0/PSI^+$) occurred sequentially in the 1990s

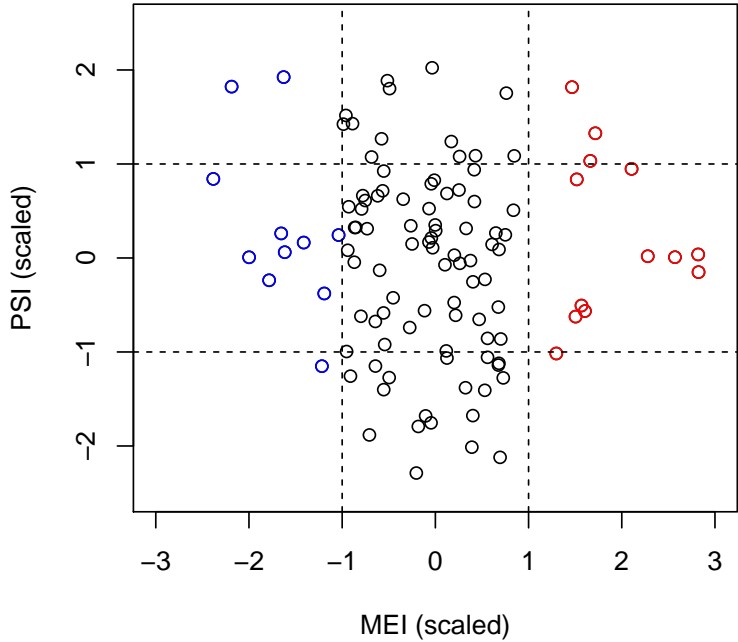

**Figure 8.** Scatter plot of the new split index (PSI) and MEI, the Multivariate ENSO Index (each normalized to unit variance). Dashed lines give (minus) one standard deviation of MEI and PSI indices, respectively. Red (blue) dots mark warm (cold) ENSO events.

| Combination | $PSI^+$ | $PSI^-$ | $PSI^0$ |
|---|---|---|---|
| $AAO^+ MEI^+$ | 3 | – | – |
| $AAO^+ MEI^-$ | 2 | – | – |
| $AAO^+ MEI^0$ | 8 | – | 5 |
| $AAO^- MEI^+$ | – | – | 1 |
| $AAO^- MEI^-$ | – | 1 | – |
| $AAO^- MEI^0$ | – | 12 | 6 |
| $AAO^0 MEI^+$ | 2 | – | 7 |
| $AAO^0 MEI^-$ | – | – | 8 |
| $AAO^0 MEI^0$ | 3 | 6 | 47 |

**Table 4.** The predominant jet regimes ($PSI^+$ = split jet, $PSI^-$ = single jet, $PSI^0$ = mixed jet regime) as counted during the 111 winter months of the 1979–2015 period, separated by occurred combinations of AAO and ENSO states.

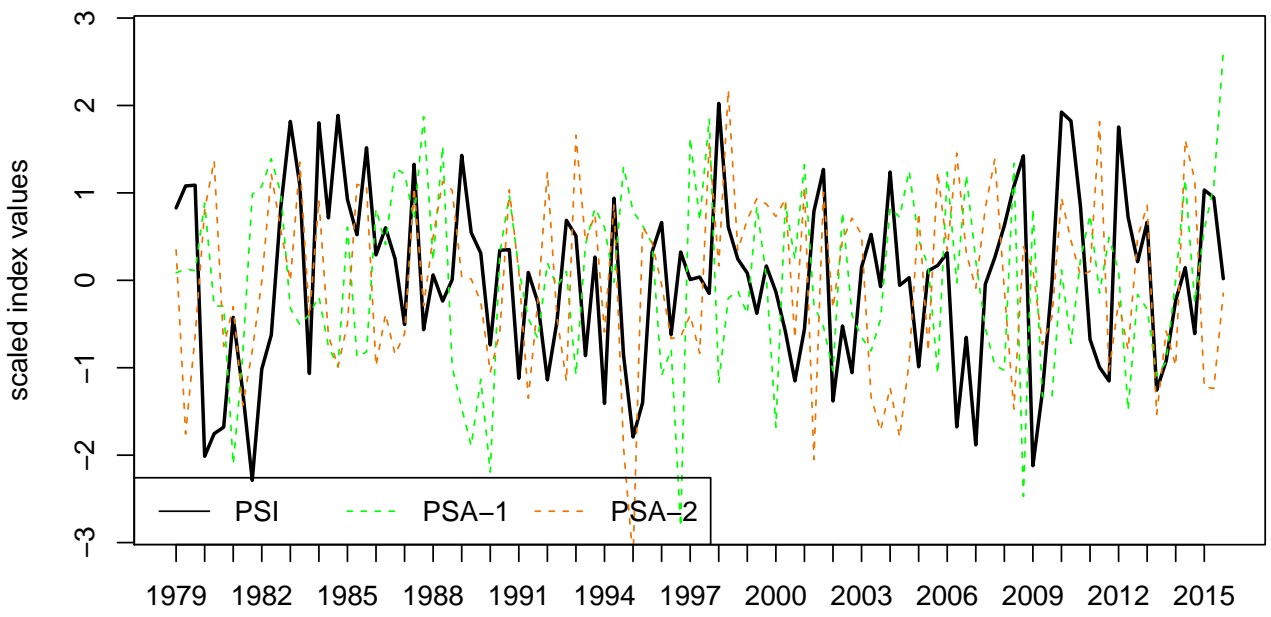

**Figure 9.** Time series of PSI (black solid line), PSA-1 index (green dotted line) and PSA-2 index (orange dotted line) from 1979 to 2015.

(not shown). A spectral analysis of the time series of the differences between PSI and AAO index shows multiple frequencies over the 37 year period (not shown). Unfortunately, the study interval is too short to draw meaningful conclusions about decadal to centennial variabilities. Furthermore, the time series of differences is correlated neither with the ENSO nor to the known Pacific Decadal Oscillation (NOAA's PDO index, not shown).

5    The low correlative relationship between MEI and PSI mentioned above arises from the fact that 8 out of 18 split events occurred during any flavor of ENSO phases, while only one cold ENSO event occurred in a negative PSI phase. The last three rows of Table 4 reveal that there are 7 (8) ENSO warm (cold) events, including the extreme La Niña event in 1988 and the severe El Niño event from 1997, where a mixed jet ($PSI^0$) regime predominated.

Figure 9 shows the time series of both the PSA modes and the PSI. Although the correlation between PSI and the PSA indices
10   is low ($r_{PSA-1} = 0.19$ and $r_{PSA-2} = 0.06$), the PSA-1 mode is significantly correlated with the individual PCs of the SH zonal circulation variability (Table 3). Both PSI components (i.e. PC2 and PC3) are associated with the PSA-1 pattern. The spatial pattern of EOF2 was previously identified as resembling the dominant part of the split jet structure (Section 3.1) characterised by an intensified PFJ over the eastern hemisphere. Since the PSA-1 pattern is primarily associated with the variability over the

central and western Pacific sectors, it can be concluded that PC2 describes less PSA-1 variability but contains a considerable part of the wind response associated with the AAO. PC3 accounts for the "non-annular" component of the PSI and contains most of the variability over the Pacific sector. However, only 3 cases were identified where both indices, PSI and PSA-1, exceeded the threshold of one standard deviation during the analysis period (not shown).

## 5  Summary and discussion

While several studies of the SH winter circulation examined the presence, origin, and structure of the climatological time-mean SH split jet which forms over the western South Pacific Ocean in the vicinity the South Australia / New Zealand region, only a few have developed indices capturing the SH split variability to quantify the relationships with the large-scale variabilities i.e. the AAO and the ENSO.

According to our literature review, until now there are three published studies (Bals-Elsholz et al. (2001); Inatsu and Hoskins (2006); Yang and Chang (2006)) suggesting a SH split jet index. All of which have used a region based definition, which contradicts the assumption that the SH split jet is a hemispheric feature rather than a regional effect. The most important advantage and the innovative aspect of our new PC based split index (PSI) compared to the existing indices is its independence from regional definitions: the PSI is defined by linearly combining the SH second and third principal component of the monthly (JAS) anomalous zonal wind at 200 hPa.

The PSI correlates well with the individual split indices and the mean of all gives a significant correlation coefficient of 0.76 with the PSI. An analysis of Bals-Elsholz et al. (2001) showed that their SFI agreed favorably with a subjective classification of the split regime with the exception of 1998. This year was affected by a strong La Niña event and an anomalously weak STJ.

While the three earlier split indices failed to agree with the (subjective) split regime, the PSI value for July 1998 was high and was thus marked as one of 18 split events. From this fact, the correlation analysis, and the investigation of PSI composites in the 200 hPa zonal wind field, it is evident that our PC based split index is able to reproduce the SH wintertime split jet structure.

Separate composites of PC2 and PC3 reveal that both PCs are essential for the split representation. PC2 is associated with the jet strength (Lorenz and Hartmann, 2001) over the eastern hemisphere, whereas PC3 exhibits a "non-annular" component in both composites. We propose that the split events are connected to PC2 because it represents the jet strength, and to PC3 as it adds a "non-annular" component to the jet variability. Thus, the PC based split jet index (PSI) is a qualitative measure of the spatio-temporal variability of the split jet and can efficiently be used to investigate the relationships with the known climate variability modes i.e. AAO, ENSO, and PSA patterns.

Although not previously reported in studies involving the SH circulation (jet) variability (e.g. Gallego et al. (2005)), it was expected that the SH split jet should exhibit a strong connection to the AAO. Our newly developed PSI shows a highly significant correlation with the leading mode in the near-surface level of the geopotential height anomalies (NOAA's AAO at 700 hPa) with r = 0.81.

This highly significant relation of the PSI to the leading mode in geopotential reiterates the importance of the AAO to the PFJ variability (Fyfe, 2003; Gallego et al., 2005). The PFJ strength is enhanced under $AAO^+$ conditions in accordance with the negative pressure anomalies over the polar region occurring under $AAO^+$. Since the PFJ variability is more important for the representation of split events (Bals-Elsholz et al., 2001) than its subtropical counterpart, the AAO has a high correlation with the PSI (r = 0.81). Nevertheless, the AAO index has a neutral value in 5 out of 18 split events occurring in the analysis period between 1979 and 2015. It can be concluded that a positive AAO phase is a preferred condition for the SH split jet. However, the split jet is also able to arise in neutral AAO months.

The correlation between the MEI and the PSI is low, but 7 out of 18 split events occurred during a warm (5) or cold (2) ENSO month. The occurrence of both warm and cold ENSO events damps the correlation with the PSI, but the relative frequency of ENSO events is still 30 %. It seems likely that any kind of ENSO flavor is favorable for split events, which results in a nonlinear relation between PSI and ENSO i.e. positive (negative) PSI values mainly occurred during warm or cold (neutral) ENSO states. From earlier studies it is known that El Niño (La Niña) phases enhance (reduce) the STJ strength over the South Pacific Ocean via advection of mean flow momentum flux (e.g. Chen et al. (1996)). This is consistent with our finding that 5 out of 18 split events occurred during a warm ENSO phase which enhances the STJ (PFJ) strength. Otherwise, there are strong ENSO events (El Niño of 1997 and La Niña in 1988) during the analysis period, which did not lead to a pronounced split event. Thus, an ENSO event alone is not able to reproduce the SH split jet variability. It is worth mentioning that published literature indicates that ENSO and AAO are also mutually dependent. There are reports that cold ENSO events are favored by positive phases of the AAO (e.g. Carvalho et al. (2005)). As these are in turn associated with split events, we expect La Niña phases to occur more often than El Niño events during $PSI^+$ phases.

In contrast, the ENSO warm phase coincided more frequently (5 months) with the high PSI (split jet) phases than its cold counterpart (2 months), although both flavors of ENSO appeared roughly equally during the considered time period of 111 winter months. We propose that the ENSO induced Rossby wave dynamics are beneficial for the SH split jet modulation, regardless of the sign of that oscillation. However, further work is needed here to clarify the complex relationship between ENSO and the SH split jet.

The traditional view suggests that the PSA patterns are part of a stationary Rossby wave train extending from the central Pacific to Argentina (e.g., Mo and Higgins, 1998). The PSA patterns, typically defined as the second and third modes of tropospheric geopotential height variability over the SH (e.g., Mo, 2000), have been attributed to ENSO (PSA-1) on interannual time scales and to the quasi-biennial component of ENSO and the Madden-Julian-Oscillation (PSA-2) respectively (Mo and Paegle, 2001).

The relationship between the PSA indices and the PSI revealed a low relationship to both indices ($r_{PSA-1} \approx 0.2$ and $r_{PSA-2}$ = 0.06). In contrast, the individual PCs of the zonal circulation at 200 hPa correlate significantly with the PSA-1 index. Positive PSA phases are associated with a strong PFJ over the western Pacific sector (not shown) and the PSA-1 mode is consistently significantly correlated with the PC3(-) component which is (by definition) related to split events. Negative PSA events are associated with an enhanced PFJ over the eastern hemisphere which partly describes the EOF2 variability. We propose that

PC3 represents both AAO and the PSA variability. However, the connection between the PSA-1 mode and the PSI is canceled out by the PC2 component, which is (as well as the PC3 component) negatively correlated with the PSA-1 mode.

We conclude that the strength of the relationship between the new PC based SH split jet index and the large-scale modes AAO and ENSO is nonlinear. The highly significant relationship with the AAO highlights the importance of the PFJ variability

which is a determining factor in the split jet formation, while the ENSO and PSA effects are complex and relatively less important.

Interestingly, the SH split jet representation is poor in the Earth System Model (not shown) of the Max-Planck-Institute of Meteorology (MPI-ESM) in the Medium resolution (Giorgetta et al., 2013). A possible reason for this deficiency could be a "too strong" modeled low-pressure system over the Amundsen Sea (Jungclaus et al., 2013) i.e. the prominent (surface)

Amundsen Sea Low (ASL). The underestimated variability in that model leads to significant differences in the representation of the leading EOF patterns at the 700 hPa geopotential height (and thus zonal wind) fields in the MPI-ESM model as compared to the ERA-Interim reanalysis (Babian et al., 2016).

A future extension of our study will involve the investigation of seasonal variations in the PSI. In the context of a positive summertime trend in the AAO due to ozone depletion, it is likely that the PSI exhibits significant trends at least on seasonal

time scales. Furthermore, the potential of a higher temporal resolved index will be explored. A daily index would be desirable to describe the processes connected to blocking activity over the Australia / New Zealand sector.

*Data availability.* The ERA-Interim reanalysis data used in this article are freely available from the ECMWF (https://www.ecmwf.int/en/research/climate-reanalysis/era-interim). The climate indices are available from NOAA (http://www.noaa.gov/).

*Acknowledgements.* The authors appreciate extensive discussion with Prof. Dr. Henning Rust on the statistical concepts used in this study

and PD Dr. Peter Névir for comments and insightful discussion on climate dynamics. Thanks are also due to two anonymous reviewers for careful reading the manuscript and for constructive comments.

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
