# Peer review of "A New Index For The Wintertime Southern Hemispheric Split Jet"

_Atmospheric Chemistry and Physics, 2017_

## Referee Comment (RC1) · Anonymous Referee #1 · 14 Nov 2017

This manuscript introduces a new index for the Southern Hemisphere (SH) winter split jet. The split jet is an important feature of SH winter climate which requires greater understanding due to its importance in understanding a globally-relevant part of the climate. Winter (July-September) monthly zonal wind data from a reanalysis dataset (ERA-Interim) is used to define an index based on principal component analysis. The rationale is that there is some disagreement between previous locally-defined split jet indices in terms of conclusions on links between split jet variability and large-scale modes of climate variability. The creation of a new hemispheric-scale index is motivated by the need to understand these disagreements.

With regard to the manuscript itself. I found it difficult to follow and have a number of concerns about the methodology and rationale which are crucial the main conclu-

sions. These are listed below and represent major comments that would need to be addressed for publication in ACP.

Major comments: I'll start with a recommendation in light of the below comments. My feeling is that this work would be easier to interpret and present clearly if the methodology and questions were reversed. By reversed I mean that the present approach of developing an EOF-based index that correlates with previous split jet indices could be reversed and instead the EOF analysis could be used to investigate in detail the linkages between existing split jet indices and large-scale patterns of atmospheric circulation variability. This could help to solve a number of issues listed below relating to the methodology and interpretation. In fact, in light of the major points below I find it difficult to see another option for raising the manuscript to the necessary standard. The main major points are as follows:

1. There are a number of concerns about the EOF/PC analysis and I remain to be convinced that this is an appropriate tool for developing an index of the split jet (but see above note about reversing the analysis / questions).

a. The rationale for the precise combination of PCs used for the PSI index is not explained and seems arbitrary (equation 5 on lin 25 of page 8). Why was this specific linear combination chosen?

b. The actual EOF patterns are not shown, which makes it very difficult to judge the relevance of these patterns for the split jet region and possible variability with other aspects of the SH climate system.

c. The vertical pressure level seems to be chosen on the basis of highest correlation with previous split jet indices rather than a specific dynamical reason. It is important to explain why 150 hPa is more appropriate than the lower 200/300 hPa levels used previously.

d. I'm not an expert on EOF analysis, but it seems problematic to me to just perform

this on winter months since presumably large jumps can occur between the end of one winter and the beginning of another that could introduce artefacts into the analysis of month-to-month variability. An explanation of why this approach is acceptable or what was done to avoid this is important.

2. The use of a combination of PC2 and PC3 used in the PSI index are potentially problematic in discussion of links with ENSO (section 4.2). The context for this is that in tropospheric geopotential height, the 2nd and 3rd PCs are associated with two variants of the so-called Pacific South American (PSA) pattern (e.g. see Cai and Watterson, 2002), which are often interpreted as representing Rossby wave trains from the tropics (e.g. Karoly, 1989).

3. The relatively low correlation between the PSI index and previous split jet indices is a concern (Table 2). Correlations of around 0.7 only explain 50% of the variance and to me this indicates that the new index could be mixing up a number of other correlated dynamical features that exist on the hemispheric scale. In particular see the above point about PSA-1 and PSA-2. It seems possible that the PSI captures some sort of PSA-related patterns, which we know are highly correlated with the split jet region (Figure 1 of Cai and Watterson).

4. Statistical significance should be included on the composite anomaly maps shown in Figures 5 and 6, otherwise it is difficult to judge which aspects of the maps to focus on.

5. The manuscript needs to be reviewed by a fluent English speaker (or alternatively one of the many available English review services) since it is not currently of a publishable standard.

Specific comments

Page 1, line 1: I suggest mentioning explicitly that the split jet is evident in mid-latitude westerly/zonal winds.

[Figure]

Page 1, line 9: Similarly, it should be mentioned here that the principle component is defined from the zonal wind field.

Page 1, line 15: "relation" -> relationship.

Page 1, line19: "an SH" -> a SH.

Page 2, line 2: "minimum wind speeds in the westerlies of the upper level flow" -> weak upper-level westerly winds

Page 2, line 9: Mention somewhere here that the AAO is now more commonly referred to as the Southern Annular Mode (SAM), since many (most?) readers will be most familiar with the SAM terminology.

Page 2, line 26: The meaning of "which confines with the relationships between" is not clear to me. This needs to be re-worded.

Page 2, lines 34-35: I disagree that inconsistent relationships indicates deficiencies. I would prefer to say that these differences need to be understood and motivate the development of an alternative index to help in this understanding. Also, the concept of a more hemispheric definition than the previous regionally-defined indices represents a key novelty in the authors' proposed index.

Page 3, lines 1-5: I'm not sure what the main point being made here is. This should either be made clearer or possibly the paragraph could be removed. Also, it important to note that the impact of stratospheric ozone depletion in the lower-troposphere is highly seasonal (mainly summer) and not significant in austral winter.

Page 3, line 10: Note that the AAO/SAM is not specifically a low-frequency mode of variability, although it will contain a component of externally-driven low-frequency variability and trends.

Page 4, line 1: "dissolved" -> resolved.

Page 4, Section 2.2: The use of different terminology in the title (EOF analysis) and

main text (principal component analysis) is confusing and should be clarified.

Page 4, line 12: I'm not an expert on EOF analysis, but it seems odd to me to just perform this on winter months since presumably large jumps can occur between the end of one winter and the beginning of another that could introduce artefacts into the analysis of month-to-month variability. An explanation of why this approach is acceptable is important.

Page 4, line18: I don't understand what is being described in the sentence starting "Afterward, the monthly . . .". Maybe this needs clarification and/or re-wording.

Page 4, line 19: "the relationship" -> the linear relationship

Page 6, line 23: "isobar" -> "pressure level"

Page 8,line 1: It seems odd to mention the PSI here before it is defined in the next section. I would prefer to keep the focus on the individual PCs in this section.

Page 8, lines 18-19. I would remove "This gives rise to the assumption" and simply state that splits are less correlated to PC1 than PC2 and 3.

Page 8, line 22: Further explanation of what is meant by a "dynamical PC based index" is needed.

Page 8, line 23: It's not clear how this particular linear combination was decided on. This needs to be explained.

Page 8, line 29: How well correlated are the various indices with each other? A correlation of around 0.7 between the PSI and these indices explains only half of the variance and therefore could potentially capture quite different aspects of variability.

Page 9, first paragraph of section 4.1. It is important to add statistical significance to the composite difference plots. Also, it would be useful for the reader to see what these composites look like for the split jet indices rather than not show them.

Page 10, main text discussion on individual PC2 and PC3 composites: Given that these show rather different composite patterns, it would be useful to give an explanation of how the combination of the two provides an appropriate PSI.

Page 12, Section 4.2: As mentioned above, I doesn't seem appropriate to categorise the AAO/SAM as a low-frequency climate mode due to the importance of jet vacillations in its existence.

Page14, discussion of ENSO links in main text: In the lower troposphere the 2nd EOF of monthly geopotential height often shows a pattern referred to as the Pacific South American Pattern (PSA). I would encourage the authors to comment on how this might relate to their results and also PC2 in zonal winds at 150 hPa.

---

## Referee Comment (RC2) · Anonymous Referee #2 · 12 Dec 2017

This manuscript proposed a new index for the wintertime Southern Hemispheric split jet based on the principal components. Compared with the existing split indexes, this new one considers the split jet as hemispheric rather than a regional feature. Further analysis indicated that the newly defined index has a strong coherence with the Antarctic Oscillation (AAO), but the split jet variability is less dependent on the phases of ENSO.

This paper is well written, and the proposed index could be used as an extra index for understanding the mechanism of the wintertime Southern Hemispheric split jet. I have major concerns regarding the ENSO modulation of the split jet variability as detailed below. At this point I cannot recommend publication of this paper.

Major point

The modulation of ENSO on the split jet index has not been robustly investigated in this study, since the time scale studied here for the split jet index is the sub-seasonal, while the ENSO varies on seasonal to inter-annual time scales. The proposed index is highly correlated with AAO, which has a strong month to month variability, so it is not surprising to see that there is no correlation between the index and ENSO on the monthly time series. I would recommend investigating the relationship between the seasonal mean split jet index and the seasonal mean ENSO index.

---

## Author Comment (AC1) · 23 Jan 2018

**Response to Referees**

**–**

**A New Index For The Wintertime Southern Hemispheric Split Jet**

S. Babian, J. Grieger, U. Cubasch

January 23, 2018

For a better readability all reviewers' comments are written in italics. The associated responses are written in roman style and are blue colored. New paragraphs, which were added to the manuscript are also written in roman style (black).

**1   Response to Referee #1**

*This manuscript introduces a new index for the Southern Hemisphere (SH) winter split jet. The split jet is an important feature of SH winter climate which requires greater understanding due to its importance in understanding a globally-relevant part of the climate. Winter (July-September) monthly zonal wind data from a reanalysis dataset (ERA-Interim) is used to define an index based on principal component analysis. The rationale is that there is some disagreement between previous locally-defined split jet indices in terms of conclusions on links between split jet variability and large-scale*

*modes of climate variability. The creation of a new hemispheric-scale index is motivated by the need to understand these disagreements. With regard to the manuscript itself. I found it difficult to follow and have a number of concerns about the methodology and rationale which are crucial the main conclusions. These are listed below and represent major comments that would need to be addressed for publication in ACP.*

*Major comments:*
*I'll start with a recommendation in light of the below comments. My feeling is that this work would be easier to interpret and present clearly if the methodology and questions were reversed. By reversed I mean that the present approach of developing an EOF-based index that correlates with previous split jet indices could be reversed and instead the EOF analysis could be used to investigate in detail the linkages between existing split jet indices and large-scale patterns of atmospheric circulation variability. This could help to solve a number of issues listed below relating to the methodology and interpretation. In fact, in light of the major points below I find it difficult to see another option for raising the manuscript to the necessary standard.*

Thanks for this recommendation. It shows that we had not sufficiently explained our line of arguments. To remedy this weakness, we revised the introduction of our manuscript and added the following paragraph:

Additionally, the literature lacks a description of whether the split jet variability is related to the Pacific South American (PSA) patterns, although these were found to be associated with both teleconnection indices, ENSO and the AAO. The PSA patterns are conventionally seen as Rossby wave trains emanating in quadrature to each other from the tropical Pacific towards Argentina and serve simultaneous as waveguide for eddies moving south. On interannual time scales the PSA-1 mode is tied to ENSO, while the PSA-2 pattern is associated with the quasi-biennial component of ENSO (e.g., Mo, 2000). Furthermore, although called an "annular" mode, the AAO contains asymmetries, which are most pronounced in Austral winter and over the Pacific sector. This (tropically forced) component of the AAO is related to a fixed active Rossby wave

source and resembles the spacial structure of the PSA patterns (Ding et al., 2012). The lack of knowledge about the relationship between the split variability and the PSA patterns and the inconsistencies among earlier studies concerning the connections to ENSO and the AAO motivate the development of an improved SH split jet index.

The underlying idea of our work is the analysis of the SH split jet variability and its links to the large-scale modes of climate variability. To archive this aim the established split jet indices were calculated. We found that they have large discrepancies among each other. The inconsistency among the three split indices stems from their rigid definitions in different meteorological fields, areas and levels and results in varying relationships to the known modes of climate variability. The authors' approach was therefore to abandon these rigid definitions and to apply the (in the climate community more common) EOF analysis for the whole SH instead in order to describe the SH split jet variability. Therefore, the EOF analysis of the 200 hPa zonal wind field is the most central aspect to define the new index. For better readability we added a paragraph at the beginning of section 3, introducing our motivation:

As mentioned previously, the wintertime split jet is the most prominent asymmetric feature of the mid-latitude SH circulation centered over the Pacific sector although it bears as well hemispheric signatures (Yang and Chang, 2006). In order to design a PC based split jet index, three earlier defined split jet indices (Section 2.4) were reproduced and their statistical relationships to the leading modes of the SH wintertime circulation (as depicted by zonal wind anomalies at 200 hPa) have been investigated. The EOF modes showing the largest coherence with the split jet indices (as defined by the respective studies) are assumed to contain the main signals associated with the split variability.

*1a) The rationale for the precise combination of PCs used for the PSI index is not explained and seems arbitrary (equation 5 on line 25 of page 8). Why was this specific linear combination chosen?*

PC2 and PC3 were chosen with respect to the correlation analysis between the earlier

split indices and the leading 5 PCs of the zonal wind field (table 2 in the manuscript), which confirms that the split variability is contained in particular these components. The fact, that all three very different indices correlate significantly with these PCs, indicates that our method is consistent and that it is appropriate to linearly combine these into a new split index. Adding PC1 to the linear combination failed to improve the correlative relations.

*1b) The actual EOF patterns are not shown, which makes it very difficult to judge the relevance of these patterns for the split jet region and possible variability with other aspects of the SH climate system.*

The negative and positive index phases of PC2 and PC3 do not mirror each other, i.e. the spatial patterns look different for positive/negative PC index composites. Since we are interested in the spatial pattern of the positive composite of PC2 and the negative of PC3, the authors decided for composite instead of regression plots. Nevertheless, we added the regression plots here (Fig.1) for comparison to the composite plots.

*1c) The vertical pressure level seems to be chosen on the basis of highest correlation with previous split jet indices rather than a specific dynamical reason. It is important to explain why 150 hPa is more appropriate than the lower 200/300 hPa levels used previously.*

The PSI was previously calculated for several vertical levels from the surface to the stratosphere, but the correlation to the mean of all earlier split indices maximizes in the 150 hPa level. Thus, and due to the mechanisms associated with the jet(s) variability, which have their maximum at that level (Akahori and Yoden, 1997), the 150 hPa level was chosen for the split definition. Fig.2 displays the correlation values between the PSI (defined in several vertical levels) and the mean of all three earlier split indices and shows that primarly the tropopause level is associated to the split variability. Since the correlation value of the PSI in the 200 hPa level of the zonal wind is as well highly significant and in order to facilitate the comparison to the older split indices (which are

defined in 200 and 300 hPa levels), the authors thank for the constructive advise and change the level of PSI definition from 150 hPa to 200 hPa.

*1d) I'm not an expert on EOF analysis, but it seems problematic to me to just perform this on winter months since presumably large jumps can occur between the end of one winter and the beginning of another that could introduce artefacts into the analysis of month-to-month variability. An explanation of why this approach is acceptable or what was done to avoid this is important.*

Indeed, using subsets of yearly time series instead of all months of the year can lead to jumps in the PC time series. This should be taken into account for considerations concerning e.g. frequency analysis. In awareness of this fact, the authors selected methods (composite and correlation analysis), which are not sensitive to these jumps. A similar procedure is applied by NOAA for monthly and seasonal EOF analysis[1]. Several authors are calculating the seasonal EOFs in the same manner as we did (e.g., Wallace et al., 1993; Baba and Renwick, 2017).

*2) The use of a combination of PC2 and PC3 used in the PSI index are potentially problematic in discussion of links with ENSO (section 4.2). The context for this is that in tropospheric geopotential height, the 2nd and 3rd PCs are associated with two variants of the so-called Pacific South American (PSA) pattern (e.g. see Cai and Watterson, 2002), which are often interpreted as representing Rossby wave trains from the tropics (e.g. Karoly (1989)).*

The calculation of the PCs which are used to define the PSI, is performed with zonal wind in 200 hPa, the PSA patterns are associated with PCs of the geopotential height in 500 hPa. An EOF mode to EOF mode connection cannot be expected for two physical related fields (Ambaum et al., 2001) and therefore no connection between PSA (in geopotential height) and PSI (in zonal wind) was expected. Nevertheless, the authors agree that the links between the leading modes of the geopotential height field and the
* * *
[1]See (https://www.esrl.noaa.gov/psd/enso/mei.ext/) and references therein.

EOF modes of the zonal wind are worthwhile (especially in context of the nonlinear linkage between split events and ENSO). To summarize, our new result indicates that the PSA-1 mode is associated with the PC3 component of the PSI (r = 0.55). However, while the correlation to the PC3 component is significant, the PSI shows weak correlation to both PSA modes ($r_{PSA-1} = 0.2$ and $r_{PSA-2} = 0.1$, respectively) – consistent with the earlier split indices. The low correlation values between PSI and PSA modes (as defined in (Mo, 2000)) were added to table 3 in the manuscript and the dynamical mechanisms are discussed and added to the last section. We thank for this comment and mention this aspect now in section 4.2.:

Fig. 9 in the manuscript shows the time series of both PSA modes and the PSI. Although the correlation between PSI and the PSA indices is low ($r_{PSA-1}$ = 0.19 and $r_{PSA-2}$ = 0.06), the PSA-1 mode is significantly correlated to the individual PCs of the SH zonal circulation variability (Tab. 1). Both PSI components (i.e. PC2 and PC3) are associated to the PSA-1 pattern. The spacial pattern of EOF2 was previously identified to resemble the dominant part of the split jet structure (Section 3.1) marked by an intensified PFJ over the eastern hemisphere. Since the PSA-1 pattern is primarily associated with the variability over the central and western Pacific sectors, it can be concluded that PC2 describes less PSA-1 variability but contains a considerable part of the wind response associated with the AAO. PC3 accounts for the "non-annular" component of the PSI and contains most variability over the Pacific sector. It is argued, that PC3 contains remarkable PSA-1 variability. However, only 3 cases were identified where both indices, PSI and PSA-1, exceeded the threshold of one standard deviation during the analysis period (not shown).

*3) The relatively low correlation between the PSI index and previous split jet indices is a concern (Table 2). Correlations of around 0.7 only explain 50% of the variance and to me this indicates that the new index could be mixing up a number of other correlated dynamical features that exist on the hemispheric scale. In particular, see the above point about PSA-1 and PSA-2. It seems possible that the PSI captures some sort of PSA-related patterns, which we know are highly correlated with the split jet region*

|  | PSI | PC2 | PC3 | BE | YC | IH | Mean |
|---|---|---|---|---|---|---|---|
| JAS – correlation values | | | | | | | |
| **AAO** | **0.81** | **0.65** | **-0.50** | **0.45** | **0.66** | **0.58** | **0.64** |
| **MEI** | -0.09 | 0.07 | 0.20 | 0.20 | 0.12 | **0.34** | **0.25** |
| **PSA-1** | 0.19 | **-0.26** | **-0.55** | 0.14 | 0.16 | -0.17 | 0.05 |
| **PSA-2** | 0.06 | 0.20 | 0.13 | 0.22 | 0.20 | -0.12 | 0.11 |

**Table 1.** Monthly (JAS) Pearson correlation coefficients of the three split indices introduced in the methods section, as well as the PSI with three major SH climate mode indices for the Antarctic Oscillation (AAO), ENSO (Multivariate ENSO Index) and the Pacific South American (PSA) patterns (PSA-1 and PSA-2 indices as defined in Mo (2000)). Bold values are significant at the $\alpha$ = 1% level.

*(Figure 1 of Cai and Watterson).*

As mentioned previously, it is not straightforward to connect EOF modes of two physical related fields to each other (see previous answer). By definition, the leading mode of the SH 500 hPa geopotential height field is the AAO (Mo, 2000, e.g.). Since the corresponding PCs are by definition uncorrelated to each other, it is not surprising that the PSI (which is significantly correlated (r= 0.81) to the AAO) does not show a statistical relationship to the PSA modes (i.e. the second and third modes of the ZG field under investigation). Nevertheless, the relationships between the earlier split indices, PSI and the PSA modes were re-examined (previous answer).

*2c) Statistical significance should be included on the composite anomaly maps shown in Figures 5 and 6, otherwise it is difficult to judge which aspects of the maps to focus on.*

Significances are now included.

*2d) The manuscript needs to be reviewed by a fluent English speaker (or alternatively one of the many available English review services) since it is not currently of a publishable standard.*

The manuscript will be reviewed by a professional native English speaker before the publication.

[Figure]

*Minor (textual) comments:*
Page 1, line 1: I suggest mentioning explicitly that the split jet is evident in mid-latitude westerly/zonal winds.
The mid-latitude westerlies are explicitly mentioned now.
Page 1, line 9: Similarly, it should be mentioned here that the principle component is defined from the zonal wind field.
The sentence was changed to: Our index is based on the principal components (PC) of the zonal wind field for the SH wintertime.
Page 1, line 15: "relation" → relationship.
The "relation" was reworded to relationship.
Page 1, line 19: "an SH" → a SH.
The "an SH" was changed to "a SH".
Page 2, line 2: "minimum wind speeds in the westerlies of the upper level flow" → weak upper-level westerly winds.
The rewording was adopted.
Page 2, line 9: Mention somewhere here that the AAO is now more commonly referred to as the Southern Annular Mode (SAM), since many (most?) readers will be most familiar with the SAM terminology.
The authors thank for the advice and added the SAM terminology.
Page 2, line 26: The meaning of "which confines with the relationships between" is not clear to me. This needs to be re-worded.
The question has been changed to: Can an index be defined which clarifies the relationships between the SH split jet and the large-scale teleconnection indices ENSO and AAO?.
Page 2, lines 34-35: I disagree that inconsistent relationships indicates deficiencies. I would prefer to say that these differences need to be understood and motivate the development of an alternative index to help in this understanding. Also, the concept of a more hemispheric definition than the previous regionally-defined indices represents a key novelty in the authors' proposed index.

The authors agree, that inconsistencies do not equal deficiencies. The sentence has thus been changed to: These inconsistencies motivate the development of an improved SH split jet index.

Page 3, lines 1-5: I'm not sure what the main point being made here is. This should either be made clearer or possibly the paragraph could be removed. Also, it important to note that the impact of stratospheric ozone depletion in the lower-troposphere is highly seasonal (mainly summer) and not significant in austral winter.

The authors agree and deleted the paragraph.

Page 3, line 10: Note that the AAO/SAM is not specifically a low-frequency mode of variability, although it will contain a component of externally-driven low-frequency variability and trends.

The authors agree that the categorization of the AAO/SAM as low-frequency mode is mistakable in that case and decided for the terminology "large-scale" mode of the SH variability.

Page 4, line 1: "dissolved" → resolved.

The rewording has been adopted.

Page 4, Section 2.2: The use of different terminology in the title (EOF analysis) and main text (principal component analysis) is confusing and should be clarified.

The authors agree and changed the sentence to: Analysis of atmospheric circulation patterns can be done by means of an Empirical Orthogonal Functions Analysis (EOF), which is also referred to as Principal Component Analysis (PCA) (Jolliffe, 2002; Hannachi et al., 2007).

Page 4, line 12: I'm not an expert on EOF analysis, but it seems odd to me to just perform this on winter months since presumably large jumps can occur between the end of one winter and the beginning of another that could introduce artefacts into the analysis of month-to-month variability. An explanation of why this approach is acceptable is important.

As mentioned in the answer to the point 1d) of the discussion, an EOF analysis is not sensible to jumps in the time series. There is a number of references, executing

monthly and seasonal EOFs in the same manner.

Page 4, line 18: I don't understand what is being described in the sentence starting "Afterward, the monthly ...". Maybe this needs clarification and/or re-wording.

The sentence was deleted since the authors decided to show the composite mean instead of the composite anomalies w.r.t. the 1979–2015 mean.

Page 4, line 19: "the relationship" → the linear relationship

The word linear has been added.

Page 6, line 23: "isobar" → "pressure level"

"isobar" has been changed to "pressure level".

Page 8,line 1: It seems odd to mention the PSI here before it is defined in the next section. I would prefer to keep the focus on the individual PCs in this section.

The authors agree and discuss the PSI in section 3.2 only.

Page 8, lines 18-19. I would remove "This gives rise to the assumption" and simply state that splits are less correlated to PC1 than PC2 and 3.

The suggestion was adopted and the sentence has been changed to: Altogether, the splits in the westerlies are less correlated to PC1, but are rather associated with the higher order PCs in the 200 hPa zonal wind field.

Page 8, line 22: Further explanation of what is meant by a "dynamical PC based index" is needed.

"Dynamical" equals here "not static", i.e. the PSI disclaims fixed geographical definitions in contrast to the earlier split jet indices. Nevertheless, the authors agree, that the word "dynamical" is confusing in that case and was thus deleted.

Page 8, line 23: It's not clear how this particular linear combination was decided on. This needs to be explained.

As stated earlier, the linear combination was chosen, due to the significant correlation values between these PCs and the earlier split indices. This fact has been highlighted now: In order to develop a PC based split jet index, these PCs, correlating well with the earlier split jet indices, are coupled to a linear combination.

Page 8, line 29: How well correlated are the various indices with each other? A correla-

tion of around 0.7 between the PSI and these indices explains only half of the variance
and therefore could potentially capture quite different aspects of variability.

PCs are by construction linear combinations of the physical field under investigation
and it is therefore not excluded that these PCs capture different aspects of the tro-
pospheric variability. However, the correlation analysis bears, that the relationships
between the large-scale variability patterns (AAO, PSA, ENSO) and the older split in-
dices are consistent with the relationships to the PSI and can thus be seen as potential
modulators of the split variability, rather than being a disturbing factor it is assumed,
that the split variability is a constructive combination of certain phases of different low-
frequency modes. The correlation value of 0.76 between the mean of the earlier indices
and the PSI captures around 60% of the variance. Differences between the PSI and
the "static" earlier indices are indicative of different input variables, levels and regional
definitions.

Page 9, first paragraph of section 4.1. It is important to add statistical significance to
the composite difference plots. Also, it would be useful for the reader to see what these
composites look like for the split jet indices rather than not show them.

The authors thank for the advice and added significances to all composite plots. The
composite analysis for the earlier split jet indices has been added here for comparison
(see figure 3). The interested reader is referred to the composite plots of Bals-Elsholz
et al. (2001), Yang and Chang (2006), Inatsu and Hoskins (2006), respectively, where
composites of the earlier split jet indices have already been published. The authors
thus disclaim to add these plots to the manuscript, since they do not add a new infor-
mation to our work.

Page 10, main text discussion on individual PC2 and PC3 composites: Given that these
show rather different composite patterns, it would be useful to give an explanation of
how the combination of the two provides an appropriate PSI.

The authors assume that the individual contribution to the spatial structure of the PSI
is questioned here. It seems that we have not well described the particular portions
of EOF2 and EOF3 patterns, therefore we added the following sentence to the ap-

propriate paragraph (p.10, l.3 ff.) to clarify this issue: In summary, the positive PSI composite benefits from the combination of the double jet structure with a pronounced minimum between the jets associated with PC2+ and the strong PFJ over the South Pacific Ocean associated with the "non-annular" component of PC3-.

Page 12, Section 4.2: As mentioned above, I doesn't seem appropriate to categorise the AAO/SAM as a low-frequency climate mode due to the importance of jet vacillations in its existence.

As already mentioned, the authors agree that the categorization of the AAO/ SAM as low-frequency mode is mistakable in that case and decided for the terminology "large-scale" mode of the SH variability.

Page 14, discussion of ENSO links in main text: In the lower troposphere the 2nd EOF of monthly geopotential height often shows a pattern referred to as the eastern Pacific South American Pattern (PSA). I would encourage the authors to comment on how this might relate to their results and also PC2 in zonal winds at 150 hPa.

The authors re-examined the relationships between split indices and the PSA patterns, defined as EOF modes (Mo, 2000). The results are summarized in table 1. It seems, that especially PC3 is associated with the PSA-1 mode of the ZG field in 500 hPa (r = 0.55). Since EOF3 bears strong variability over the South Pacific sector, where the PFJ shows large variance, the authors conclude that the split jet resembles parts of the spatial structure of the PSA-1 mode. The re-examined table 1 was therefore added to the manuscript in Section 4.2. as well as to the discussion in section 5:

The traditional perception suggests that the PSA patterns are part of a stationary Rossby wave train extending from the central Pacific to Argentina (e.g., Mo and Higgins, 1998). The PSA patterns, typically defined as $2^{nd}$ and $3^{rd}$ modes of tropospheric geopotential height variability over the SH (e.g., Mo, 2000), have been attributed to ENSO (PSA-1) on interannual time scales and to the quasi-biennial component of ENSO and the Madden-Julian-Oscillation (PSA-2), respectively (Mo and Paegle, 2001). The relationship between the PSA indices and the PSI revealed a low relationship to both indices ($r_{PSA-1} \approx 0.2$ and $r_{PSA-2} = 0.06$). In contrast, the individual PCs of

the zonal circulation in 200 hPa correlate significantly to the PSA-1 index. Positive PSA phases are associated with a strong PFJ over the western Pacific sector (not shown) and the PSA-1 mode is consistently significantly correlated to the PC3(-) component, which is (by definition) related to split events. Negative PSA events are associated with an enhanced PFJ over the eastern hemisphere, which represents the partly EOF2 variability. It is reasoned that PC3 represents both, AAO as well as PSA variability. However, the connection between the PSA-1 mode and the PSI is canceled out by the PC2 component, which is (as well as the PC3 component) negatively correlated to the PSA-1 mode.

**References**

Akahori, K. and Yoden, S.: Zonal Flow Vacillation and Bimodality of Baroclinic Eddy Life Cycles in a Simple Global Circulation Model, Journal of the Atmospheric Sciences, 54, 2349–2361, 1997.

Ambaum, M. H. P., Hoskins, B. J., and Stephenson, D. B.: Arctic Oscillation or North Atlantic Oscillation?, Journal of Climate, 14, 3495–3507, 2001.

Baba, K. and Renwick, J.: Aspects of intraseasonal variability of Antarctic sea ice in austral winter related to ENSO and SAM events, Journal of Glaciology, 63, 838–846, doi:10.1017/jog.2017.49, 2017.

Bals-Elsholz, T. M., Atallah, E. H., Bosart, L. F., Wasula, T. a., Cempa, M. J., and Lupo, a. R.: The wintertime southern hemisphere split jet: Structure, variability, and evolution, Journal of Climate, 14, 4191–4215, doi:10.1175/1520-0442(2001)014<4191:TWSHSJ>2.0.CO;2, 2001.

Ding, Q., Steig, E. J., Battisti, D. S., and Wallace, J. M.: Influence of the Tropics on the Southern Annular Mode, Journal of Climate, 25, 6330–6348, doi:10.1175/JCLI-D-11-00523.1, https://doi.org/10.1175/JCLI-D-11-00523.1, 2012.

Hannachi, A., Jolliffe, I. T., and Stephenson, D. B.: Empirical orthogonal functions and related techniques in atmospheric science: A review, Int. J. Climatol., 27, 1119–1152, 2007.

Inatsu, M. and Hoskins, B. J.: The seasonal and wintertime interannual variability of the split

jet and the storm-track activity minimum near New Zealand, Journal of the Meteorological Society of Japan, 84, 433–445, doi:10.2151/jmsj.84.433, 2006.

Jolliffe, I. T.: Principal Component Analysis, Springer Series in Statistics, Springer, New York, 3rd edn., 2002.

Karoly, D. J.: Southern hemisphere circulation features associated with El Niño-Southern Oscillation events, Journal of Climate, 2, 1239–1252, 1989.

Mo, K. C.: Relationships between Low-Frequency Variability in the Southern Hemisphere and Sea Surface Temperature Anomalies, J. Climate, 13, 3599–3610, 2000.

Mo, K. C. and Higgins, R. W.: The Pacific–South American Modes and Tropical Convection during the Southern Hemisphere Winter, Monthly Weather Review, 126, 1581–1596, doi:10.1175/1520-0493(1998)126<1581:TPSAMA>2.0.CO;2, https://doi.org/10.1175/1520-0493(1998)126<1581:TPSAMA>2.0.CO;2, 1998.

Mo, K. C. and Paegle, J. N.: The Pacific–South American modes and their downstream effects, International Journal of Climatology, 21, 1211–1229, doi:10.1002/joc.685, http://dx.doi.org/10.1002/joc.685, 2001.

Wallace, J. M., Zhang, Y., and Lau, K.-H.: Structure and Seasonality of Interannual and Interdecadal Variability of the Geopotential Height and Temperature Fields in the Northern Hemisphere Troposphere, Journal of Climate, 6, 2063–2082, doi:10.1175/1520-0442(1993)006<2063:SASOIA>2.0.CO;2, 1993.

Yang, X. and Chang, E. K. M.: Variability of the Southern Hemisphere Winter Split Flow – A Case of Two-Way Reinforcement between Mean Flow and Eddy Anomalies, Journal of Atmospheric Sciences, 63, 634–650, doi:10.1175/JAS3643.1, 2006.

---

## Author Comment (AC2) · 23 Jan 2018

For a better readability all reviewers' comments are written in italics. The associated responses are written in roman style and are blue colored. New paragraphs, which were added to the manuscript are also written in roman style (black).

**1   Response to Referee #2**

*This manuscript proposed a new index for the wintertime Southern Hemispheric split jet based on the principal components. Compared with the existing split indexes, this new one considers the split jet as hemispheric rather than a regional feature. Fur-*

[Figure]

*ther analysis indicated that the newly defined index has a strong coherence with the Antarctic Oscillation (AAO), but the split jet variability is less dependent on the phases of ENSO. This paper is well written, and the proposed index could be used as an extra index for understanding the mechanism of the wintertime Southern Hemispheric split jet. I have major concerns regarding the ENSO modulation of the split jet variability as detailed below. At this point I cannot recommend publication of this paper.*

*Major comment:*
*The modulation of ENSO on the split jet index has not been investigated in great detail in this study, since the time scale studied here for the split jet index is the sub-seasonal, while the ENSO varies on seasonal to inter-annual time scales. The proposed index is highly correlated with AAO, which has a strong month to month variability, so it is not surprising to see that there is no correlation between the index and ENSO on the monthly time series. I would recommend investigating the relationship between the seasonal mean split jet index and the seasonal mean ENSO index.*

The relationship between the split jet variability and ENSO is very complex and changing in time. The missing correlation between PSI and MEI stems from the fact, that both La Niña as well as El Niño events are associated with a strengthening of one of the jets (PFJ or STJ). This asymmetric response to the ENSO leads thus to a low and insignificant correlation to the MEI. The positive correlation during strong La Niña phases cancels the negative correlation during El Niño events out. Thus, the time series do not correlate on a significant level. Fig.4 shows further that the relationship is changing in time as well. That does not mean that there is no relationship between the split jet variability and ENSO, rather the true interdependency between both is hidden.